# Plant-Based Formulas and Liquid Feedings for Infants and Toddlers

**DOI:** 10.3390/nu13114026

**Published:** 2021-11-11

**Authors:** Yvan Vandenplas, Nele De Mulder, Elisabeth De Greef, Koen Huysentruyt

**Affiliations:** Pediatric Gastroenterology, KidZ Health Castle, UZ Brussel, Vrije Universiteit Brussel, 1090 Brussels, Belgium; nele.demulder@uzbrussel.be (N.D.M.); elisabeth.degreef@uzbrussel.be (E.D.G.); koen.huysentruyt@uzbrussel.be (K.H.)

**Keywords:** toddler, growing-up milk, plant-based drink, plant-based beverage, plant-based formula, young child formula

## Abstract

Exclusive breastfeeding is the recommended feeding for all infants. Recent research has focused on the importance of balanced feeding during the first 1000 days, starting at conception with a balanced diet of the pregnant woman, up to the age of two years. The following step, a balanced diet after the age of two years is a challenge, as the dietary intake becomes more diversified. The role of young-child formula in this process is debated. This paper discusses the use of planted-based drinks, since they are a valuable and progressively more popular alternative for cow’s milk, if nutritionally adapted to the requirements of toddlers. Plant-based drinks are per definition lactose free.

## 1. Introduction: Drink, Beverage, Liquid Feeding: What’s in a Name?

Soy and hydrolyzed rice infant formulas exist on the market since many years. The popularity of soy formula is decreasing, despite recent expert opinions stating that soy protein isolate formula is safe, affordable, and an alternative option for cow’s milk-based formula for term infants [1]. On the opposite, hydrolyzed rice based infant formula is being used more and more, especially in infants with digestive difficulties such as functional gastro-intestinal disorders or when cow’s milk allergy is suspected [2]. Plant based infant formulas have been evaluated extensively and have been shown to be nutritionally adequate, and will be discussed more in depth in this paper [1,2,3].

However, what about balanced feeding after the age of one or two years old? Are plant-based feedings for toddlers a nutritionally adequate option? According to international recommendations toddlers should drink 300 to 500 mL milk per day [4]. Cow’s milk-based growing up milks are commercially available since many years. The European Food Safety Authority (EFSA) published nutritional requirements for children 1–3 years old [4]. The composition of some of them is listed in Table 1.

Due to an increasing market demand, more plant-based drinks have become available for toddlers. This evolution also creates a semantic dilemma: what is the best name for these plant-based products? There is consensus that the word “milk” should be reserved for the milk produced by mothers and animals, resulting in mother’s milk, cow’s milk, goat’s milk, horse’s milk, etc. As a consequence, plant-based drinks cannot be named “milk”: soy formula or soy drink should be used, not soy milk.

Plant based products for toddlers need to be divided in two clearly separated groups: those products that are nutritionally adapted to the nutritional needs of young children and those that are nutritionally insufficient. In general: the ‘drink’ category can be used by young children, but the composition of the many of these is nutritionally inappropriate and should not be considered “a feeding”.

In Belgium, cow’s milk remains one of the most important energy and protein sources for the majority of toddlers [5]. Up to now, the situation in Belgium—as in most European countries—was relatively simple: besides “soy drinks for toddlers” there was only rice-based hydrolyzed infant formula on the market. Both are nutritionally adapted for infants, toddlers, and young children, respectively.

Nowadays more nutritionally adapted plant-based products for toddlers become available. If all plant-based products are designated as “drink”, the health care provider as well as the consumer will become confused, since some are nutritionally adapted and others not (Table 2). Therefore, we propose to consider a different terminology to distinguish these two groups: “drink” or “beverage” for the nutritionally un-adapted products and “plant based liquid feeding” for those nutritionally adapted.

## 2. Soy Based Infant Formula and Drinks

Soy has been cultivated since the 17th century and is the third most important crop after rice and corn. Soy infant formula is historically the first plant-based feeding for infants and young children that was developed. The first use of soy based infant feeding was reported in the United States in 1909 [6]. Since the mid-1960s, the composition of soy based infant formula was adapted to the nutritional needs of infants [4]. Based on the National Health and Nutrition Examination Survey 2003–2010, 12% of infants aged 0–12 months old in the United States consume soy based infant formula [7]. The market share of soy based infant formula in New Zealand and Israel was reported to be 13% and 31.5%, respectively [8,9]. In 1929, soy based infant formula was introduced as a management option for infants with cow’s milk allergy (CMA) [10]. Currently, this recommendation is abolished, and soy infant formula is no longer recommended as a first-choice option in the management of infants with CMA, although often proposed as second option in many guidelines [11].

Soybean is a good source of protein. Protein content in raw soybean is around 40%, and is higher than other macronutrient content such as carbohydrate (25%) and fat (20%). Different from regular soy beverages that use raw soybean as its ingredient, soy based infant formula uses soy isolate protein. Soy isolate protein is processed from soybean by eliminating other nutrients to yield a high purity protein product that contains at least 90% of protein and maximum 1% of fat, 0.2% of crude fiber, 4% of carbohydrate [12,13]. Heat treatment and extraction during the processes are also lowering some unfavorable ingredients, such as isoflavones, trypsin inhibitor, phytic acid content and the beany flavor [14,15]. Soy protein isolate has a protein digestibility-corrected amino acid score (PDCAAS) of one which is comparable to high quality animal protein such as casein and egg white [16]. Natural fiber in soy, such as raffinose and stachyose, can cause infants to have unpleasant semiliquid stools or flatulence [14]; therefore, removing natural fiber during the process to produce soy isolate protein could be advantageous. To fortify soy protein isolate formula, the use of other types of fiber, such as fructooligosaccharides, which have a prebiotic effect, is more beneficial [17]. Fructo-oligosaccharides originate from plants and consequently do not contain traces of milk as might occur with galactooligosacharide originating from cow’s milk. The prebiotic effect helps maintain a healthy gastrointestinal environment by selectively stimulating the growth of nonpathogenic bacteria, such as bifidobacterial, and increase the frequency of defecation and decrease stool consistency [18,19,20,21].

Nevertheless, the remaining uncertainties regarding isoflavones, the phytoestrogen content, the presence of a trypsin inhibitor and the beany flavor made soy infant formula disappear from the European market. However, soy drinks for toddlers remain available, and are frequently used in toddlers and children with lactose intolerance. Besides availability and palatability, personal beliefs, religious background, and contemporary views advocating the importance of plant-based food in the adult population also influence the nutritional choices parents make for their offspring [22]. Soy protein, and thus with phytoestrogens, and soy isolate protein, and thus without phytoestrogens, are both used in toddler’s soy drinks. Therefore, while some soy-drinks on the market are not nutritionally adapted to the needs of toddlers, others are adapted, creating confusion for the health care provider and consumer, indicating the need for a different name and classification.

## 3. Rice Based Infant Formula and Drinks

Worldwide, rice is the most cultivated crop. Health care providers and parents are familiar with hydrolyzed rice-based infant formula for the treatment of CMA. Hydrolyzed rice infant formulas are present in many European countries since more than 30 years, and occupy a significant market share [23]. The consumption of hydrolyzed rice protein formulas accounted in 2018 in France for almost 5% of all formulas for children aged 0–3 years [24]. Initially this rice-based hydrolysate was significantly cheaper than cow’s milk based extensive hydrolysates, but this difference in cost has disappeared in many countries, while it persisted in others.

Hydrolyzed rice formula was shown to be safe, nutritionally adequate and tolerated by infants allergic to cow’s milk and soy formula [20,22,23]. Infants fed a rice hydrolysate for 6 months were shown to have normal growth, normal plasma biochemical levels, and no adverse reactions were seen. Hydrolyzed rice infant formula was proposed to be more effective than soy and comparable to a casein hydrolysate in infants with CMA [25]. No significant differences between a rice hydrolysate, soy infant formula and extensive casein hydrolysate groups were observed for the z-score of weight for age during the first two years of life and for nutritional serum parameters such as plasma total protein, albumin, prealbumin, calcium, magnesium, and alkaline phosphatase [25,26,27,28].

Unlike the rice-protein based infant formula, rice beverages made from rice flour are nutritionally inadequate for infants. The amino-acid lysine is deficient in rice protein, and is supplemented in rice based infant formula. The high content of arsenic in some rice cereals and other rice products received a lot of attention [29,30]. The Food and Drug Administration in the USA and the Nutrition Committee of ESPGHAN warned also about the high arsenic content in some rice products [30,31]. However, the arsenic content in rice formula was reported to be very low, and not different from the arsenic content of cow’s milk based infant formula [32].

Today, the conclusion about rice hydrolysate infant formula does not differ a lot from the conclusion from the “*Diagnosis and Rationale for Action against Cow’s Milk Allergy*” (DRACMA) guidelines published in 2010: rice protein based infant formula may provide a potentially adequate alternative if standard milk- or soy protein-based formulas are not tolerated [33]. Therefore, we endorse the conclusions of the recently published position paper of the Committee on Nutrition of the French Society of Pediatrics (CNSFP): Hydrolyzed rice protein formulas are proposed as a plant-based alternative to cow’s milk protein-based extensively hydrolyzed formulas (CMP-eHF) [34]. Hydrolyzed rice protein formulas do not contain phytoestrogens and are derived from non-genetically modified rice [2,35]. Hydrolyzed rice formula ensures satisfactory growth from the first weeks of life for infants and toddlers, both in healthy children and in those with CMPA, and they can be used to treat children with CMA either straightaway or in second intention in cases of poor tolerance to CMP-eHF for organoleptic reasons or for lack of efficacy [36].

Last but not least, cost and acceptability should as well be considered. Overall, rice hydrolysates are reported to have a better taste, although this has not been scientifically validated. In countries were CMP-eHFs are not reimbursed by the National Health System, cost of formula does play a major role in determining the final choice. A recent consensus paper positioned hydrolyzed rice formula as an alternative to CMP-eHF as first option in the dietary management of infants with CMA in the Middle East, North Africa, and Pakistan Region [2].

## 4. Other Plant Based Drinks and Formula

Although the market share of soy-based formula and drinks decreases, there is a growing demand from consumers for plant based liquid feedings for toddlers as an alternative to dairy. However, many of the plant-based beverages are not nutritionally adapted for the needs of infants and toddlers. This is a potentially dangerous evolution, because offering drinks to young children not fulfilling their nutritional requirements will result in malnutrition and failure to thrive as long-term negative consequences [3]. Therefore, the development of nutritionally adapted plant-based liquid feedings for toddlers is a welcomed and needed evolution.

More information on the concentration of individual amino acids, poly-unsaturated fatty acids, minerals, trace elements, vitamins in the majority of these plant-based drinks for toddlers are needed to draw appropriate conclusions. For many products only general data on macronutrients are available.

Plant based toddlers liquid feedings based on almond and buckwheat are available in some countries and online. A nutritionally adapted young child formula exists based on almond, buckwheat, and tapioca. Compared to other products, who are mainly based on a combination of isolated or fractionated nutrients, (i.e., combination of a protein ingredient, a fat ingredient, and a carbohydrate ingredient separately), this product uses whole foods, where nutrients are in their food matrix and the proteins are intact, minimizing the need of food processing [35].

Almonds are known to be a good protein source (around 25% of total energy content), containing monosaturated fatty acids (around 50% of total energy content), and dietary fiber (insoluble/soluble fiber at 4:1). Almond protein contains all essential amino acids with good digestibility, but sulfur amino acids (methionine + cysteine) are the first limiting amino acid, followed by lysine and threonine, in total seed proteins [36]. Almonds are tree nuts and thus one of the major allergens, but almond allergy in young children is lower compared to cow milk or soy allergy. Anaphylaxis due to almond allergy is very rare for this age group [37,38]. The potential of using almonds for feeding young children with CMA was shown in a study in infants aged 5 to 9 months with documented CMA fed almond formula demonstrated similar growth to those fed soy-based formula or the CMP-eHF [39]. Supplementation with a fortified almond formula did not cause secondary sensitization, which was observed in the other two groups [39]. However, there are insufficient data available with this type of plant-based feeding to be recommend in children with CMA as allergy to the plant protein may increase with increased use [40].

Buckwheat is recognized as a good source of nutritionally valuable protein, carbohydrate, and dietary fiber, and has received increasing attention as a potential functional food. Buckwheat seeds contain 100 to 125 mg/g of proteins, 650 to 750 mg/g of starch, 20 to 25 mg/g of fat, and 20 to 25 mg/g of mineral. The protein content in buckwheat flour is lower compared to oat flour, but significantly higher than that in rice, wheat, millet, sorghum, maize, etc. Buckwheat is not only a good source of protein in terms of quantity, but it is also a complete protein, containing all nine of the essential amino acids, which are not produced naturally and should be supplemented in humans through food [41]. Whereas buckwheat is recommended to be one the first solid foods to introduce in allergic infants mainly in Japan and Korea [42], and a popular nutrient in baby food products; it is very rarely used in Western countries where only case reports are published [43].

In Belgium, a rice-pea based nutritionally adapted young child formula has become commercially available (Table 2). While rice is used here for the carbohydrate content, pea (Pisum sativum) is used as protein source. Methionine and cysteine levels are too low in pea protein, and need to be supplemented. Pea protein is also used since many years in animal feeding [44]. A pea-based drink for toddlers seems a real novelty, but a PubMed search shows a first paper in Spanish from 1953 using pea flour as a base for infant formula [45]. A trial performed in 1959 in South Africa showed that the protein value of a maize/pea vegetable mixture was nutritionally adequate for pre-school children [46]. Pea protein isolate is available on the market since many years, and is consumed by many athletes to improve their performance. Pea protein is very soluble, and therefore, easy digestible and absorbed. The PDCAAS of the pea-protein isolate is 89% (in comparison: that of soy isolate is 92%) [47,48]. Pea protein is rich in branched chain amino acids, arginine, lysine, and iron. A so called “oat based” liquid feeding for toddlers is similar, as it uses oat as the carbohydrate source but pea as the protein source. Some of these plant-based liquid feeding for toddlers are quite expensive (Table 2).

Chick-pea alone or supplemented with methionine was shown to be a suitable milk substitute in malnourished children with lactose intolerance [49]. The percentage of absorption, retention and biological value of the chick-pea formula were 72.4, 26.4 and 35.1, respectively and 69.6, 24.3 and 34.0 in the same order, with a soy formula [50]. Since the nutritional quality of the chick-pea was not different to the commercial soy formula and the diarrhea was better controlled by the former, this formula could be recommended in the treatment of breastfed babies with lactose intolerance [50]. A clinical trial was performed 20 years ago with pea protein infant formula, but was tested in young women [51]. The results confirmed the inhibitory and enhancing effects of phytic acid and ascorbic acid respectively on iron absorption, but also indicate a relatively high fractional iron absorption from the pea-protein-based formulas. After adjusting for differences in iron status, data indicate that iron absorption from dephytinised pea protein might be less inhibitory than from dephytinised soybean protein [51]. Extruded chickpea and yellow pea protein had good indispensable amino acid digestibility in moderately stunted children, which was 20% higher than an earlier report of their digestibility when pressure-cooked, measured by the same method in adults [52]. Higher digestibility of lysine and proline highlights better retention of these amino acids in chickpea during extrusion-based processing [53].

There is little allergy reported to pea protein, and is therefore, marked as a protein source for individuals with CMA. However, pea protein is not an-allergic since a Canadian case-series has been published on anaphylaxis to hidden pea protein [54]. Evidence for sensitization to peptides present in Pisum sativum was low but requires further verification with regard to conformational epitopes [54]. In peanut-sensitive patients, cross-allergenicity was demonstrated to be most marked between the extracts of peanut, garden pea, chick pea, and soybean [55]. The results have important implications for selection of effective hypoallergenic diets and for the diagnosis of patients hypersensitive to foods [55]. The pea-rice toddlers liquid feeding does not contain sweeteners; a milk flavoring agent was added to make this drink taste as milk. Consumers invoke frequently the following reasons regarding pea and rice plant-based nutritionally adapted feedings: 100% plant based (sustainability considerations), absence of major allergens, absence of lactose, offer dietary variety.

## 5. Conclusions

Plant-based drinks that are nutritionally adapted to the requirements of toddlers are a suitable alternative for cow’s milk to ensure a balanced dietary intake in toddlers.

## Figures and Tables

**Table 1 nutrients-13-04026-t001:** Composition of some cow’s milk based growing up milk or young child formulas.

	EFSA-Recom FUF Min-Max	Whole CM	Bambix GUM 1+	Nan Optipro 3	Nan Optipro 1+	Nutrilon GUM 1+ (Eazypack)	Kruidvat 4
En (kcal/100 mL)	60–70	64	56	67	66	65	68
Lipid (g/100 kcal)	4.4–6	5.6	4.3	4.5	4.7	4.0	4.0
Sat (g/100 kcal)	NR	3.3	0.4	0.9	1.2	1.3	1.6
Unsat (g/100 kcal)	NR	2.3	3.9	3.6	3.5	2.7	2.4
LinolAn-6 (g/100 kcal)	0.5–1.2	0.1	1.1			0.5	0.5
CH (g/100 kcal)	9–14	7.3 (Lact)	13.4 (3.6 Lact + 9.8 MD)	13.4 (Lact)	13.0 (10.9 Lact, 2.1 corn starch)	13.4 (9.4 Lact, 7.7 mg inositol)	13.1 (9.1 Lact, 3.4 MD)
Fibre (g/100 kcal)	NR	0.0	0.0	0.0	38 mg 2′FL	1.4 GOS/FOS	0.9
Protein (g/100 kcal)	1.6–2.5	5.2	2.1	1.5	1,5	2.0	2.2
Ca(mg/100 kcal)	50	188	161	190	190	188	89
Na (mg/100 kcal)	25	69	60	20	55	40	
K (mg/100 kcal)	80	244	161		/	232	101
Fe (mg/100 kcal)	0.6	0.3	2.9	1.8	1.8	1.8	1.8
Vit D (µg/100 kcal)	2	0.6	3.8	1.6	1.5	4.8	2.5
Vit B12 (mg/100 kcal)	0.1	0.7	0.5	0.6	0.17	0.6	0.28
Liquid/powder		Liquid	Liquid	Powder	Liquid	Powder	Powder
Aroma		−	+	−	−	+	−
€/litre		0.89	1.71	2.48	2.09	2.03	1.4

Legend; EFSA; European Food Safety Authority; FUF: follow-up formula; recom: recommendation; min: minimum; max: maximum; CM: cow’s milk GUM: growing-up milk; YCF: young child formula; En: energy; Sat: saturated; Unsat: unsaturated; LinolA: linoleic acid; CH: carbohydrate; Lact: lactose; MD: maltodextrins; Dex: dextrose; Malt: maltose; gluc: glucose; fruct: fructose; Ca: calcium; Na: natrium (sodium); K: kalium (potassium); Fe: iron; Vit: vitamin; FOS: fructo-oligosaccharide; GOS: galacto-oligosaccharide; 2’-FL: 2-fucosyllactose; “+”: aroma added; “−”: no aroma added.

**Table 2 nutrients-13-04026-t002:** Composition of some plant-based drinks and liquid feeding for toddlers.

	EFSA-Recom FUF Min-Max	Whole CM	Bambix Rice Drink	Bambix Soy	Else Toddler	Premeriz 3	Alpro YCF 1–3+	Alpro Oat GUM 1–3+	Alpro Not Milk 3.5%	Bébé Mandorlé 3	Novalac Novarice
En (kcal/100 mL)	60–70	64	56	59	67	65	64	60	59	65	66
Lipid (g/100 kcal)	4.4–6	5.6	4.3	4.1	5.0	4.3	3.3	5.5	5.9	4.6	5.0
Sat (g/100 kcal)	NR	3.3	0.4	0.3	0.55	1.4	0.5	0.5	0.7	1.3	2.0
Unsat (g/100 kcal)	NR	2.3	3.9	3.7	4.45	2.9	2.8	5.0	5.2	3.3	3.0
LinolA n-6(g/100 kcal)	0.5–1.2	0.1	1.1	1.0		0.9				0.8	0.9
CH (g/100 kcal)	9–14	7.3 (Lact)	13.4	13.2	10.6	11.7	13.0	9.8	9.7	11.9	11.2
Fibre (g/100 kcal)	NR	0.0	0.9	0.8	1.0	0.6 FOS	0.6	0.3	1.7	0.3	0
Protein (g/100 kcal)	1.6–2.5	5.2	2.1	2.0	2.8	2.3	3.9	3.0	1.2	3.0	2.7
Calcium (mg/100 kcal)	50	188	161	161	122	124	188	200	203	163	92
Na (mg/100 kcal)	25	69	60	56	31	31	25	73	81	31	41
K (mg/100 kcal)	80	244	161	153	172	145				34	102
Fe (mg/100 kcal)	0.6	0.3	2.9	2.7	2.2	1.5	3.3	2.3		1.4	1.2
Vit D (µg/100 kcal)	2	0.6	3.6	3.4	1.7	2.7	2.3	2.5	1.3	2.1	1.5
Vit B12 (mg/100 kcal)	0.1	0.7	0.5	0.5	0.3		0.6	0.6	0.6	0.3	0.3
Liquid/powder		Liquid	Liquid	Liquid	Powder	Powder	Liquid	Liquid	Liquid	Powder	Powder
Aroma		−	+	+	−	−	+	+	−	−	−
€/litre		0.89	1.79	1.71	6.64	1.25	1.55	2.49	2.09	5.93	5.90

Legend; EFSA: European Food Safety Authority; FUF: follow-up formula;; recom: recommendation; min: minmum; max: maximum; CM: cow’s milk; GUM: growing-up milk; YCF: young child formula; En: energy; Sat: saturated; Unsat: unsaturated; LinolA: linoleic acid; CH: carbohydrate; Lact: lactose; MD: maltodextrins; Dex: dextrose; Malt: maltose; gluc: glucose; fruct: fructose; Ca: calcium; Na: natrium (sodium); K: kalium (potassium); Fe: iron; Vit: vitamin; FOS: fructo-oligosaccharide; “+”: aroma added; “−”: no aroma added.

## Data Availability

Not applicable.

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
