# Peer review of "Plant-Based Formulas and Liquid Feedings for Infants and Toddlers"

_nutrients, 2021, doi:10.3390/nu13114026_

Round 1

Reviewer 1 Report

This manuscript “Plant-based liquid feedings for toddlers” aims to describe the role of plant based drinks and formulas in the toddler.   This is a very important and timely topic, with the increased use of plant based products and helps identify the multiple gaps in understanding and areas of confusion.  Additional clarifications would be helpful in enhancing the value of the manuscript: 

Major Revisions:

  1. The aim of the manuscript is to "discuss the use of plant-based drinks for toddlers”.  There are multiple areas where the use of plant based infant formulas are discussed.  Plant based infant formulas is an important topic but seems very different from the nutritional needs of a toddler.  Would recommend either redefining goals of the manuscript and adding additional discussion about infant plant based nutrition or removing the discussion of plant based infant formulas.
  2. The authors identify the total volume of milk needed per day as 300 to 500 ml for a typical toddler but would suggest adding to this section an overview of the macronutrient and micronutrient needs for toddlers, to use as a comparison throughout the manuscript.
  3. Consider adding section identifying areas for future research.

Minor Revisions:

  1. The table is a great addition to the manuscript. Would recommend reorganizing so that it is more clear which are formulas and which are drinks/beverages.

Author Response

Reviewer 1

This manuscript “Plant-based liquid feedings for toddlers” aims to describe the role of plant based drinks and formulas in the toddler.   This is a very important and timely topic, with the increased use of plant based products and helps identify the multiple gaps in understanding and areas of confusion.  Additional clarifications would be helpful in enhancing the value of the manuscript:

We thank the reviewer for the nice comment

The aim of the manuscript is to "discuss the use of plant-based drinks for toddlers”.  There are multiple areas where the use of plant based infant formulas are discussed.  Plant based infant formulas is an important topic but seems very different from the nutritional needs of a toddler.  Would recommend either redefining goals of the manuscript and adding additional discussion about infant plant based nutrition or removing the discussion of plant based infant formulas.

We adapted the title, and the goal of the manuscript, since we think it makes sense to continue the discussion on plant-based "feedings" from infancy into toddlerhood. The introduction was also adapted.

1The authors identify the total volume of milk needed per day as 300 to 500 ml for a typical toddler but would suggest adding to this section an overview of the macronutrient and micronutrient needs for toddlers, to use as a comparison throughout the manuscript.

We thank the reviewer for this suggestion. The lay-out of the table was adapted, and theoretical needs for macro- and some micronutrients of toddlers are also listed in the table. 

Consider adding section identifying areas for future research.

Done

Reviewer 2 Report

Here is my opinion about the paper titled: Plant-based liquid feedings for toddlers. Authors compared children milk to non-milk drink (based mainly on rise and soy). 

  1. The paper is not a "article" type (there was no materials and methods section) but a review type.
  2. Authors had a conflict of interest – they compared milk with non-milk drinks and took honoraria’s for the Companies from the food sector.
  3. Authors propose a terminology to distinguish two groups: "drink" or "beverage" for the nutritionally un-adapted products and "plant based formula" for those nutritionally adapted. The authors short described based formula and drinks from soy, rice and other plant but did not specify exactly which products are suitable for toddlers and which are not.
  4. Table 1 is nowhere referred in the text. Moreover, the table is hardly legible and poorly discussed - and it is only table in the paper! It's hard to know what drinks are being described (which products listed in the table are toddler formulas and which are beverages, and what plants were used to produce).
  5. Authors should described detailed information on the studied products, e.g. concentration of individual amino acids, elements, PUFA composition, etc., for appropriate conclusions drawing. Based on very general data (protein / fat / sugar content), they cannot answer the question posed in the hypothesis.

There are also several minor faults, such as: wrong order of publication in the references (number 20 should be 19, number 21 should be 20 etc.), some abbreviations are not explained, eg Pis s 1, etc.

Author Response

Reviewer 2

Authors propose a terminology to distinguish two groups: "drink" or "beverage" for the nutritionally un-adapted products and "plant based formula" for those nutritionally adapted. The authors short described based formula and drinks from soy, rice and other plant but did not specify exactly which products are suitable for toddlers and which are not.

Table 1 is nowhere referred in the text. Moreover, the table is hardly legible and poorly discussed - and it is only table in the paper! It's hard to know what drinks are being described (which products listed in the table are toddler formulas and which are beverages, and what plants were used to produce).

Authors should described detailed information on the studied products, e.g. concentration of individual amino acids, elements, PUFA composition, etc., for appropriate conclusions drawing. Based on very general data (protein / fat / sugar content), they cannot answer the question posed in the hypothesis.

There are also several minor faults, such as: wrong order of publication in the references (number 20 should be 19, number 21 should be 20 etc.), some abbreviations are not explained, eg Pis s 1, etc.

Round 2

Reviewer 1 Report

Authors addressed suggestions posed by this reviewer.  I recommend accepting the manuscript in its current format.

Reviewer 2 Report

The authors did not answer directly to my comments. The paper, however is currently more informative, than before. About the conflict of interest the editors should make this decision.